# Drivers, Risk Factors and Dynamics of African Swine Fever Outbreaks, Southern Highlands, Tanzania

**DOI:** 10.3390/pathogens9030155

**Published:** 2020-02-25

**Authors:** Folorunso O. Fasina, Henry Kissinga, Fredy Mlowe, Samora Mshang’a, Benedict Matogo, Abnery Mrema, Adam Mhagama, Selemani Makungu, Niwael Mtui-Malamsha, Raphael Sallu, Gerald Misinzo, Bishop Magidanga, Fredrick Kivaria, Charles Bebay, Solomon Nong’ona, Fred Kafeero, Hezron Nonga

**Affiliations:** 1Food and Agriculture Organization of the United Nations, Dar es Salaam 14111, Tanzaniaraphael.sallu@fao.org (R.S.); fred.kafeero@fao.org (F.K.); 2Zonal Veterinary Center, South West Zone, Sumbawanga 55101, Tanzania; henry.kissinga@yahoo.com; 3District Veterinary Office, Ileje District Council, Ileje 53205, Tanzania; mlowefredy@gmail.com; 4Department of Livestock and Fisheries, Mbeya District Council, Mbeya 53101, Tanzania; dr.samoraone1@yahoo.com; 5District Veterinary Office, Chunya District Council, Chunya 53535, Tanzania; bbemack@gmail.com; 6Tanzania Veterinary Laboratory Agency Zonal laboratory, Iringa 51101, Tanzania; mremaabnery@gmail.com; 7Office of the Regional Administrative Secretary, Mbeya Region, Mbeya 53101, Tanzania; adamhagama@gmail.com; 8Directorate of Veterinary Services, Ministry of Livestock and Fisheries, Dodoma 41000, Tanzania; makungu57@gmail.com (S.M.); nongahezron@yahoo.co.uk (H.N.); 9SACIDS Foundation for One Health, Sokoine University of Agriculture, Morogoro 67000, Tanzania; 10Centre for Infectious Diseases and Biotechnology, Tanzania Veterinary Laboratory Agency, Dar es Salaam 15487, Tanzania; bishopignas@gmail.com; 11Food and Agriculture Organization of the United Nations, ECTAD Regional Office for Eastern Africa, Nairobi 00100, Kenya; Fredrick.kivaria@fao.org (F.K.); charles.bebay@fao.org (C.B.); 12Veterinary Investigation Center, Iringa 51101, Tanzania

**Keywords:** participatory epidemiology, participatory disease surveillance, African swine fever, risk factors, disease drivers, disease dynamics, livelihood, rural economy, Tanzania

## Abstract

African swine fever remains an important pig disease globally in view of its rapid spread, economic impacts and food implications, with no option of vaccination or treatment. The Southern Highlands zone of Tanzania, an important pig-producing hub in East Africa, is endemic with African swine fever (ASF). From approximately the year 2010, the recurrence of outbreaks has been observed and it has now become a predictable pattern. We conducted exploratory participatory epidemiology and participatory disease surveillance in the Southern Highlands to understand the pig sector and the drivers and facilitators of infections, risk factors and dynamics of ASF in this important pig-producing area. Pigs continue to play a major role in rural livelihoods in the Southern Highlands and pork is a major animal protein source. Outbreaks of diseases, particularly ASF, have continued to militate against the scaling up of pig operations in the Southern Highlands. Intra- and inter-district and trans-border transnational outbreaks of ASF, the most common disease in the Southern Highlands, continue to occur. Trade and marketing systems, management systems, and lack of biosecurity, as well as anthropogenic (human) issues, animals and fomites, were identified as risk factors and facilitators of ASF infection. Changes in human behavior and communication in trade and marketing systems in the value chain, biosecurity and pig management practices are warranted. Relevant training must be implemented alongside the launch of the national ASF control strategy for Tanzania, which already established a roadmap for combating ASF in Tanzania. The high-risk points (slaughter slabs, border areas, and farms with poor biosecurity) and high-risk period (November–March) along the pig value chain must be targeted as critical control points for interventions in order to reduce the burden of infection.

## 1. Introduction

Globally, African swine fever (ASF) has become a disease of great concern, with threats to international trade and food security, especially with the recent expansion of geographical territories to cover previously unreached parts of the world [1,2]. In 2019 alone, naïve ASF-free countries of China, Cambodia, Mongolia, Democratic People’s Republic of Korea, the Republic of Korea, Lao People’s Democratic Republic, Indonesia, Myanmar, Philippines, Timor-Leste and Vietnam in Asia and newer territories in Europe have been infected [2,3] and countries that are free are at constant threat of infection [1,2,3,4]. ASF is endemic in many African countries and Sardinia, Italy, and it continues to infect and re-infect the countries of Africa [1,4,5,6,7,8,9]. ASF is caused by the ASF virus (ASFV), a large complex icosahedral DNA virus and the only member of the Asfarviridae family, genus Asfivirus whose length can range from approximately 170 to 193 kbp [1,4,5]. The disease may manifest in per-acute, acute, subacute or chronic forms and ASF is usually characterized by mild to severe lesions and clinical signs including febrile conditions, erythema and cyanosis of the skin followed by high mortality rates of up to 100% depending on the form of infection [2,4,5]. ASF can manifest in four epidemiological cycles including (1) the ancient tick bite sylvatic cycle, (2) the tick–pig cycle, (3) the domestic (pig–pig) cycle, and (4) the wild boar–habitat cycle [1,2,3,4,5,6]. The European wild boar (*Sus scrofa*) and feral pigs are very susceptible to ASF like the domestic pigs but the Warthog (*Phacochoerus africanus*), the red river hog (*Potamochoerus porcus*) and bushpig (*Potamochoerus larvatus*) usually remain asymptomatic [2,3,4,5,6].

The United Republic of Tanzania experienced outbreaks of African swine fever (ASF) since its first occurrence in the country in 1962 [10]. Major epidemics of ASF were reported in 1987 and 1988 in the regions of Mbeya and Arusha/Kilimanjaro, respectively, with significant economic losses and impacts on food and nutritional securities. Since 2010, ASF outbreaks resurged in at least 15 local municipalities of Tanzania, spreading across wide areas of Tanzania, particularly in the Southern Highlands; to date, the Mbeya and Songwe Regions have been declared endemic for ASF despite intense efforts at control [10]. Detailed findings of these previous outbreaks are available in published reports [10,11,12,13,14,15,16,17].

From 2016, ASF appeared to have assumed a predictable annual cyclical pattern of infections in different locations in the Southern Highlands, Tanzania, a situation that has forced stakeholders in the pig value chain to adopt modified practices, perceptions, attitudes and behaviors, including the entrenchment of ‘knowledge’ of ASF. Primarily, genotypes IX, X, XV and XVI have circulated widely until 2010, when a virulent genotype II caused outbreak in the Southern Highlands with near 100% mortality [11,12]. The understanding of spread and spatial patterns and risk factors for infection and spread have been confirmed as necessary for disease prevention and control [2,18]. Such understanding and comprehensive analysis of the risk variables, drivers of infection and dynamics that intensify the recurrence of outbreaks of ASF from the stakeholders’ viewpoints are critical to support efforts at controlling and eradicating localized infection, like the one in the Southern Highlands. The objective of the current study was to evaluate the gaps identified above using both qualitative and semi quantitative epidemiological tools and provide guidance on the control of ASF in Tanzania. The outcome of these analyses should positively impact on rural livelihoods and economies, as well as benefit food and nutritional security in Tanzania, particularly in the Southern Highlands. This will also have positive ramifications through lessons learnt for countries with similar socio-economic status and livestock systems.

## 2. Materials and Methods 

This work was conducted with the permission of the Director of Veterinary Services, Ministry of Livestock and Fisheries, Dodoma, Tanzania, under the approval number: MA 154/355/16. All respondents were willingly recruited and were made aware of their rights and privileges before enrolment for participation. Each participant was free to discontinue participation at any stage of the process without a notification. All efforts were made to comply with biosecurity protocols and ensured that no farm was contaminated in the process of conducting this study.

### 2.1. Review of Recent Outbreaks

A desk review of all outbreak reports and recent ASF outbreaks including rumors of outbreaks in the Southern Highlands was undertaken. We received outbreak reports from Veterinary and Livestock Field Officers responsible for the infected and non-infected selected wards. To qualify for inclusion, the village must have been located in one of the districts in the Southern Highlands. The Veterinary and Livestock Field Officers were probed through in-depth questions where details in the reports need clarity and all information obtained were confirmed with the Directorate of Veterinary Services, Ministry of Livestock and Fisheries, Dodoma, Tanzania. Epidemiologic data collected include timelines of outbreaks and rumors, historical outbreaks, formal and anecdotal reports, field interventions, control/eradication efforts, livestock markets and movement patterns. 

### 2.2. Identification of Locations, Stakeholders, Development of Tools and Interviews

The Veterinary and Livestock Field Officers from selected villages were pre-informed of the planned activities and such officers organized the randomly selected stakeholders for interview (focus group discussions (FGDs) and in-depth interviews (IDIs)). The FGDs, guided by checklist, and the IDIs, developed as categorical questions, were administered to selected stakeholders. Data for the FGDs were collected continuously until the saturation point was reached (i.e., no new issue was raised by participants). The use of categorical data for the IDI was to ease data processing, minimize variation and improve response precision. While the FGDs extended between two and three hours, the IDIs lasted for approximately an hour. 

Pre-selected veterinarians were trained on the use of participatory epidemiology and participatory disease surveillance (PE/PDS) using available training manuals [19,20,21]. A list of previous risk factor variables for the introduction and transmission of ASF were harmonized from various reports [14,15,16,18,22]. These variables were transformed to narrative questions to stimulate discussions for PE/PDS. Narratives questions were pre-tested and modified to aid clarity during field applications. Field officers (n = 7) who administered the questionnaire were trained in the use of the developed questions. Emphases were placed on the PE/PDS rules to avoid: 1) leading the participants to answers, 2) domination of focus group discussion by few individuals, 3) inadvertent introduction of expert opinions on the participants and 4) avoidance of other factors that may cause bias in the use of PE/PDS tools. Using focus group discussions (FGDs) (n = 11), community-related variables were obtained on livestock management systems, major activities and occupation in the village, trades and transport system in livestock, outbreaks of ASF, reporting system, practices before, during and after outbreaks or rumor of outbreak of ASF, and general narrative associated with livestock in the community. Seasonal calendars were used among the communal groups to obtain timelines on diseases and likely annual cycles of occurrence based on their experience. A similar calendar was presented to the traders, butchers and transporter group to obtain timelines on determinants of pig sales and movements. To accept an opinion or perception for the seasonal calendar, consensus was either obtained from all members of the group (100% acceptance) or where there was dissenting opinion(s), ballot was conducted and at least 90% concurrence was accepted as representative. 

In-depth interviews (IDIs) (n = 37) were conducted with randomly selected farmers, butchers, traders and other suitable stakeholders. These interviews were conducted to triangulate the FGDs, to fill missing gaps arising from the FGDs, and to collect detailed information on the biosecurity and preventative practices at household and community levels. Proportional piling (for details, see Section 2.3 below) was used to obtain consensus agreement on perceived livestock population ranks, contributions to rural household income and livelihoods, common pig diseases, as well as associated morbidity and mortalities. All tools were utilized by adopting previously standardized methods [19,20,21].

### 2.3. Proportional Piling and Consensus Mapping

Proportional piling was conducted as described in previous works [19,20,21]. Briefly, stakeholders were requested to list and write down all livestock species in each village under consideration. A total of 100 cowpeas were given to each group to divide them by proportions according to weighted numbers per 100 animals in the village, and for economic contributions using household incomes. Each group was allowed to readjust the distributions until all members were in agreement with the proportional representation. A similar process was repeated for listed diseases, morbidities and mortalities, impact of diseases on incomes and frequencies of occurrence of diseases. 

To determine the consensus map for the pig value chain in the Southern Highlands, details from the interviews conducted during FGDs, IDIs and consultations with the Veterinary and Livestock Field Officers were used to cross-validate the information. Geo-coordinates (Latitudes and Longitudes) of the surveyed locations and all other locations mentioned in the discussions were obtained. Information obtained include the following: year of outbreaks, length of outbreaks, regions, districts and wards affected, number of affected animals, number dead, likely source of infections, primary and secondary roads, major livestock (pig) markets, major water bodies (streams, rivers, lakes), pig population densities, and any other epidemiological details necessary for the outbreak and risk map. All details were submitted to the Geographic Information System (GIS) laboratory, Institute of Resource Assessment (IRA), University of Dar es Salaam, for mapping and outputs were cross verified by field veterinarians.

### 2.4. Virus Confirmation

During the course of the study, and based on the community (participants’) description of the ASF clinical signs and syndrome (anorexia, febrile conditions, erythema, cyanosis of the skin, high mortality rates of up to 100% and abortion), a farm that was experiencing the described clinical signs and syndrome was visited and sampled. A clinically sick but pregnant pig was purchased and euthanised by exsanguination. Spleen, mesenteric and gastro-hepatic lymph nodes, visceral fluid, fetal fluids, uterine fluids, spleen, serum and whole blood from the euthanized pregnant pig; and spleen from a diseased pig during the previous outbreak were collected for ASF confirmation using molecular methods. Samples were aliquoted in the Tanzania Veterinary Laboratory Agency (TVLA) Zonal laboratory in Iringa and dispatched separately to the ASF laboratory in Sokoine University of Agriculture (for partial amplification of major structural protein *VP72* gene of ASFV using PCR as described by Bastos et al. [5]), with duplicate samples sent to the Centre for Infectious Diseases and Biotechnology, TVLA, Dar es Salaam, for quantitative evaluation of the virus using reverse-transcription PCR (RT-PCR). Diagnosis was carried out using the method of Bastos et al. as described by Misinzo et al. [5,12] and Chang’a et al. [16].

### 2.5. Statistical Analysis

Descriptive statistics (mean percentages and standard deviations with 95% confidence intervals) were determined for all the quantitative values from the community interviews (FGDs), the stakeholders IDIs and details from the proportional piles.

## 3. Results

We interviewed a total of 95 individual stakeholders in 11 FGDs and 37 IDIs, covering six villages and a public slaughter facility for pigs in two districts of Mbeya and Chunya. Approximately 56.8% of the participants were male and 43.2% were female. Women participated equally with men and moderators of each FGD ensured that crowding-out, drowning of minority opinion and other cause of biases were prevented. Specifically, in 36.4% of the groups, women were the lead respondents and approximately 50% of the veterinary/livestock officers were female. Women dominated 54.5% of the groups by population and one group was almost exclusively women. Where some members of a group appeared to dominate responses, convenient control was brought in by redirection of questions to subdued group members, or discussions were initiated from the apparently weaker group members. Stakeholders were farmers, butchers, slaughter assistants, transporters, traders, and middlemen. The livestock field officers served as opinion shapers but did not influence the exact responses from stakeholders. While five of the communities have experienced previous outbreaks of ASF (2016–2017), one had an ongoing outbreaks (November 2019) and only one of the communities has never experienced outbreaks of ASF based on history and collected anecdotal evidence. Although it was indicated that pigs are kept indoors, during the post-harvesting period of crops, boars are allowed to feed in grains and other crop fields. We also observed a few roaming pigs, especially around the slaughter slabs and near the dump sites. 

### 3.1. Historical Perspectives and Spatio-Temporal Distributions of Outbreaks of ASF in the Southern Highlands, Tanzania

In the Southern Highlands, between 2010 and 2011, outbreaks of ASF were first reported in Kyela district of Mbeya region, which shares international boundaries with Malawi. Legal and illegal trades of livestock occur across the borders and may have facilitated the introduction and transmission. These outbreaks spread inwards and covered seven districts of Mbeya region (Rungwe, Ileje, Mbarali, Mbeya Rural, Chunya, Mbozi and the City of Mbeya), and in other districts including but not limited to Temeke, Ilala, and Kinondoni municipalities of Dar es Salaam. Details are available in previous reports [10] (Figure 1 and Figure 2). Chunya district had reported outbreaks in 2011, 2012, 2014, 2017 and recently in 2019 (Figure 1 and Figure 2). Mbeya district first had outbreaks in early 2017, and again in November/December 2017. These outbreaks coincided with peaks of rains in the districts and because the district has a major livestock market (Pig Market) in Chang’ombe village, which serves pig farmers from other districts, outbreaks in the district were likely disseminated to other districts inadvertently by long-distant transportation and pig trades (Figure 1 and Figure 2).

In the Rukwa region, outbreaks have been reported and sustained in 2011, 2014 and 2017. Both home slaughter and the use of slaughter slabs prevailed in Rukwa and Katavi and it is not yet known whether the ongoing outbreaks of 2019 involves the Rukwa region (Figure 1 and Figure 2). In Ileje district, outbreaks have been reported, first in November 2010, possibly from virus introduction from Mbeya Rural, which occurred at the same period with infections of ASF in Kyela, a boundary district with Malawi. The Kyela outbreaks were associated with outbreaks in Karonga and Chitipa districts of Malawi which shared trade, market access and management system with districts in the Southern Highlands zone of Tanzania. Other outbreaks have been reported in 2011, 2016 and 2017 (Figure 1 and Figure 2). Details of these reports are available in the supplementary notes and in other reports. While the 2019 outbreaks have been confirmed in Chunya, it is suspected that other districts, particularly Songwe and Tunduma, may have ongoing infections.

### 3.2. Proportional Piles for Livestock Populations, Economic Contributions and Pig Diseases in the Southern Highlands, Tanzania

A total of 13 animal species featured in the description of commonly found animals at the village level in the Southern Highlands including chickens, cattle, pigs, ducks, goats, cats, dogs, rabbits, sheep, pigeons/doves, guinea fowls, donkeys and guinea pigs. Although there were village by village differences in the ranking of animal populations by proportions in the villages, overall, out of every 100 animals, chickens ranked the highest (29 ± 20) with cattle (21 ± 8), pigs (14 ± 6) and goats (14 ± 5) following in that order. Guinea pigs are the least common animals (Table 1). In terms of economic contribution to household, rural economy and livelihoods by percentages, pigs were mentioned as contributing the greatest proportion of household incomes (40 ± 18), with poultry (21 ± 11) and cattle (19 ± 8) following. The lowest contributors to rural livelihoods in the Southern Highlands are the guinea fowls and guinea pigs (Table 1). Although other animal species are managed by the stakeholders, pig remains a preferred livestock because it is considered a fast cash earner compared to other species; however, women respondents often emphasized chickens as the most important contributors to rural household incomes during the interviews.

The common diseases and syndromes were ranked based on impacts and importance to the economy as follows: (i) African swine fever (69%), (ii) diarrhea ((7%), (iii) mange (10%), (iv) worms (10%), (v) respiratory infections (1%), vi) tremor (0.2%), (vii) abortion (2%), (viii) cyst (1%), and (ix) vomiting (1%) (Table 2, Figure 3). Similar rankings were obtained for importance based on impact of morbidity/mortality, wherein ASF ranked highest with 74% and tremor ranked the lowest with a score of 0.2% (Table 2, Figure 3). Out of every 100 pigs in the Southern Highlands, the stakeholders perceived that approximately 27 will likely become sick—of which, almost 20 will die due to ASF alone (Figure 3). However, based on frequency of occurrence per annum, worm infections (63 ± 30%) and mange (25 ± 25%) ranked highest, with ASF among the least common diseases (1.5 ± 2%) (Table 2). In one particular village, the cause of economic losses next to ASF was Cysticercus cellulosae cyst due to total carcass condemnations.

### 3.3. Determinants of Trade, Movements and Drivers of Risks of ASF Based on Perceptions and Practices in the Southern Highlands, Tanzania

The seasonal calendar indicated that most of the pig-related diseases and syndromes (mange, worms, and diarrhea among others) occurred all the year round (Figure 4), whereas ASF appears to occur cyclically and prevails from November to the first quarter of the succeeding year. By consensus, mange, worms, diarrhea ranked the highest for occurrence, reiterating the earlier quantitative analysis in Table 2. The determinants of pig sale and random movements were the prevailing pig price (whole animal), pork sale, volume of slaughter, supply-demand dynamics to the abattoir and the cropping season. The period of high ASF prevalence, November to the following March, corresponded with periods of low pig price, pork sale (except in December), volume of slaughter and supplies to the abattoir (Figure 5). The months of April to August appear to be high seasons for all the indicators above and aligned with the harvesting seasons for most crop products (Figure 5).

### 3.4. Virus Confirmation

There was a 100% match between signs and syndromes of ASF, as described by the stakeholders and molecular diagnostics based on partial amplification of the ASFV *VP72* gene by PCR and RT-PCR, indicating that the farmers’ knowledge of the syndromes and clinical signs associated with ASF is extremely good. Interestingly, samples of fetal tissues and fluids that were aseptically collected during sampling were positive for ASF, indicating transplacental transmission. All the nine samples tested positive by PCR and RT-PCR (Figure 6). All samples were found positive by PCR amplification of the *VP72* gene, with the highest positives (lowest Ct values, close to 20), recorded for spleen, whole blood and serum.

### 3.5. Qualitative Evaluations

The information detailed below arose from the qualitative interviews conducted during the FGDs and the IDIs, backed up by key informants (veterinary and livestock field officers) where detailed information was lacking. The details represent the opinions of the stakeholders and not those of the interviewers or authors.

#### 3.5.1. Biosecurity Practices at the Slaughtering Facilities and Farm Levels, Including Case Detection, Notification, Restrictions and Waste Disposal

In the Southern Highlands, farmers keep mixed breeds of pigs and villages respond to rumors of fatal disease of pigs promptly. Specifically, pig owners mostly sell off all adult pigs and retain only the piglets, young weaners and pregnant sows. Live pigs were sold to traders (farm-gate buyers) who collect pigs in the farms and slaughter at the village level or in other distant locations, and villagers may also slaughter few pigs. Most of the pig producers refused to scale-up their production system due to fear of annual cycles of ASF infection. Sometimes, farmers with infected herds refused to report to veterinary authorities but simply discarded infected dead carcasses in the community refuse dumps where scavengers (primarily dogs and other pigs) accessed the carcasses. Carcasses buried in shallow pits were sometimes dug up and feasted on by scavengers. To date, the government has no compensation policy for losses associated with ASF infections in farms and, coupled with the fact that there is no treatment for the disease, the farmers’ motivation to report outbreaks is low. 

While pig viscera (stomach, intestines, liver, spleen, etc.) and trotters were sold for local consumption or given to the slaughter assistants for use, other waste material like intestinal contents and scalded materials are dumped in the open refuse site. Due to water scarcity at the rural communities, common water sources (streams, pools and wells) were preferably used in livestock production, both for drinking and cleaning and the community believed that ASF outbreaks were associated with rains which washed infections from upstream areas downstream, or possibly transmitted through the air.

A single farmer who operated strict compartmentalization, maintained a close herd in an infected zone, and refused visits from visitors and farm-gate buyers had kept pigs since 2003, with no outbreak to date. Similarly, a community in Chunya, located at an altitude 1700 m, which used both intensive and extensive (scavenging) management systems had incurred economic losses in 2017 associated with outbreaks in other communities but has been free from infection. The significant difference in this community was in the restrictions of all cadres of stakeholders from farm visits during the outbreak period and pigs were slaughtered and shared as pork in the community, but no pork was allowed from outside the community.

Overall, biosecurity practices across the surveyed villages were poor. All farms sold their products to farm-gate buyers and raised mixed pig stocks. Approximately 68% and 81% notified neighbors and Veterinary and Livestock Field Officers during ASF outbreaks respectively. A total of 95% preferred communal sale of pig visceral or sharing it with neighbors and only 14% buried gut contents. Only 10% of the farmers implemented any form of quarantine when new pigs were purchased and 19% practiced safe disposal of waste, even though 81% removed manure routinely from the pig pen. Finally, respondents claimed to lock pens (95%), assessed pig health using professionals (92%), not mix age (84%) and not mix species (95%) (Table 3).

#### 3.5.2. Pig Trade, Livestock Markets, Livestock Auctions, Transportation and Slaughter Facilities

To provide a summarized overview, the results of the qualitative interviews for this section are as follows:(i)Movements of live pigs towards markets sites and around these sites: The identified livestock (pig) markets (Mbuyuni, Mjele (Chang’ombe village), Usangu and Uyole) were listed in the consensus map (Figure 1). While some districts have resident livestock markets located within them, some other districts share livestock markets with neighboring districts. For example, the Mbuyuni livestock (pig) market in Songwe district sources pigs from around Mbeya region (comprising of seven districts), operates twice in a month with the sale of 70–100 pigs per market day, serves and distributes pigs to many neighboring districts. Many pigs, bought from different homesteads, may be transported together to the markets in the same vehicles (pick-up vans or small buses) either as single species or sometimes mixed with goats and sheep. If any pigs die during transport, they are salvage-slaughtered, and the intestines are thrown away. To avoid quarantine, and particularly, during outbreak periods, a lot of illegal movement of live pigs and meat from infected to uninfected area occurred. Meat are transported locally using pick-up vehicles and motorcycles and, for long-distant travel, buyers may transport pigs during the night or slaughter and package and have previously put the well-wrapped meat in the petrol tank for transport to avoid detection. Previous arrest linked with such illegal transport had a confirmed distance covered of over 800 km (Mbeya to Dar es Salaam);(ii)Movements of live pigs towards slaughtering sites and around these sites: Pig traders and middlemen travel around to buy pigs in bulk from multiple sources, farms and locations (villages, wards and districts). They enter many farms directly to buy pigs and move live pigs from farms to markets and vice versa. Slaughter slabs are quite common, and most villages have privately or public owned pig slaughter facilities. Farmers sometimes bring pigs to the abattoirs/slaughter facilities unsolicited, and if such pigs were not selected for slaughter, they were kept in the human habitations near the slabs as temporary holding grounds, until the next few days when they were returned home. Scavenging pigs may also be seen around the slaughter slab (Appendix A), with implication for disease transmission;(iii)Movements of pig products from the slaughtering sites (including wastewaters): Purchased pigs can stay up to ten days or more before slaughter depending on the market dynamics. Quite a number of the pig traders also keep pigs at home, and same people handled the home-kept and purchased pigs in terms of feeding and cleaning. During outbreaks and rumors of outbreaks, cheaper prices are offered for sick and recumbent pigs. The farm-gate buyers admitted that they were aware of their contributions to introduction and transmission of infections through transportation of meat using motorcycles and small pick-up vans. They confirmed that they move from slaughter facilities to farms, often with contaminated knives and materials, ready for emergency slaughter, and without any form of disinfection. Public slaughter slabs may have pipe-borne water and supervising veterinarians or veterinary assistants, yet the wastewaters used for washing of carcasses and intestines during slaughter sometimes flowed into the holding tanks and overflowed to contaminate nearby streams (Appendix A). Such streams serve most of the communities downstream;(iv)Movements of people into the slaughtering sites and from the slaughtering sites: People sighted in the vicinity of slaughter slabs include the following: veterinary officers, veterinary assistants, butchers, slaughter assistants, middlemen, traders, farmers, farm-gate buyers, transporters and women who trades in trotters and cook pork. Butchers sometimes sell meat directly to end users (customers) who took them for roasting at a special meat grilling area called “Kitimoto” or homeward for cooking. The intestine, head and trotters are often sold to women who prepared and sold them with the local beer. Such butchers started the day by slaughtering and distributed meat portions to the Kitimoto selling point (mostly pubs) and later went home to clean and feed their own pigs.

In recent times, traders, transporters and butchers avoid buying pigs in outbreak areas and this action prevented further outbreaks in the 2017 events. The noticed behavioral change was primarily because many of the customers in the area, who also keep pigs, avoided buying pork during outbreaks. Hence, the traders, transporters and butchers were forced to comply with supplying meat from non-outbreak locations to avoid further transmission of infections. The traders were however unaware of the recent outbreaks and suspected cases in Chunya district.

#### 3.5.3. Identified Risk Factors and Facilitators of ASF Infections and Transmission

The identified risk factors and facilitators can be grouped as follows: a) Humans and anthropogenic issues (middlemen, traders, transporters, pig keepers who visit each other and sometimes, the livestock field officers);b) Fomites (vehicle, slaughter knives, motorcycles for pork transport, vehicle for moving live animals, clothes, shared equipment and facilities, streams and rivers which serve as common water source for drinking for livestock, slaughter facilities and accommodation around the slaughter slabs);c) Animals (shared boars for mating, free-roaming pigs, pigs presented at the abattoir, slaughter slabs or markets and returned home afterwards and other scavengers like dogs and cats).

## 4. Discussion

Approximately 46%–60% of all Tanzania’s pigs are found in the Southern Highlands [23,24,25]. Pig densities in the districts of the Southern Highlands range from 2 to 6 pigs/km^2^ [23]. The approximate standing pig populations for Mbeya, Iringa, Ruvuma, Njombe, Songwe and Rukwa regions are 713,063, 102,259, 222,420, 96,324, 71,245 and 80,780, respectively [25,26]. Details of the pig production systems are available in the works of Wilson and Swai [23], and those of Kimbi et al. [24]. 

The present work has shed light on the dynamics of African swine fever in the Southern Highlands, Tanzania. For almost a decade (2010–2019), the ASF virus strains inflicted untold economic hardship and decimated livelihoods with severe food security implications. In this work, the understanding of the drivers, risk factors and dynamics of ASF virus circulation in the Southern Highlands from the perspective of the often neglected, yet strategic stakeholders are very important. We interviewed a panel of stakeholders (farmers, butchers, slaughter assistants, transporters, traders, and middlemen) with a view to gain insights into the pig value chain, socio-economics, behavioral (knowledge, attitudes, practices) and anthropological issues that continue to drive and facilitate outbreaks of ASF in the Southern Highlands.

Stakeholders in the Southern Highlands considered pigs as very important contributors to rural livelihoods. Although chickens and cattle were present in greater proportions in the Southern Highlands, the economic contributions of pigs cannot be underestimated. Previous workers have confirmed the predominant role of pigs in rural livelihoods in the Southern Highlands [23,25,27]. In view of these factors, the impacts of ASF have ramifications beyond food and nutritional security alone. The farmers confirmed the willingness to scale-up pig production except that the limitations of ASF outbreaks have prevented them. A careful analysis of the work of Kimbi and colleagues [23] revealed that while other categories of livestock species have been scaled up, pig populations will appear to be stagnant or dropping in these communities. Presumably, pigs and the sale of pork are primarily the responsibilities of men, whereas chickens are seen as women’s works and a source of family income. This might have influenced the decisions of some of the women who insisted that chickens possibly play more of a role than pigs as sources of family income.

ASF was designated as the highest ranked disease by importance and impacts on morbidity and mortality (Figure 2a,b), yet it was one of the lowest ranked frequently occurring diseases. The impact of ASF was ranked at least seven times compared to the next ranked disease, this emphasized stakeholders’ perceptions and views on ASF. It should be known that the burden of worms and mange were quite heavy in terms of frequency of occurrence, and cysticercosis appeared to be an emerging problem in some localities. Any pig herd health program planned for the Southern Highlands should be comprehensive enough to consider the inclusion of these identified disease burdens. To date, there is no treatment or preventive vaccination for ASF despite a recent attempt on an experimental vaccine [28,29,30]. Hence, biosecurity remains the critical element to reduce the burden of ASF [28]. 

In the present study, the overall biosecurity systems for the pig value chain in the Southern Highlands were poor. A lot of biosecurity gaps were identified in our analysis and this should warrant the attention of veterinary authorities to safeguard the pig sector of the livestock industry, particularly in the Southern Highlands. Key training on good farming practices, biosecurity principles, risk reduction protocols and safe trade in livestock and its products must be planned and implemented in the Southern Highlands to reduce the burdens of ASF [6,18,31]. Such training is now planned and foreseen to occur synchronously with the launch of the ‘*United Republic of Tanzania’s National African swine fever Strategy and Control Plan, 2019*’ in February 2020. 

Interestingly, the stakeholders were experienced enough to know all the syndromic signs and symptoms associated with ASF and all the nine samples collected based on their descriptions were matched 100 using PCR. As such, the surveillance system (both passive and active) can take advantage of stakeholder knowledge to fast track disease diagnosis through prompt reporting. Their modified behaviors on issues of biosecurity, management practices and sales/marketing can be utilized as an entry point to drive home empirical-based interventions [31]. Of particular interest were samples from fetal fluids/tissues as well as the uterine fluid, which were positive for ASF using PCR. While Schlafer and Mebus [32] have confirmed the presence of ASF in fetal placental, amniotic fluid and fetal heart blood in an experimental infection. We confirmed similar findings under field situations [33] and this extends the frontiers for studies on ASF infection, pathology and pathogenesis. 

In addition, although the stakeholders reported that it will appear that less pathogenic strains of ASF virus are in circulation, this will need further evaluation as 1) pathology may be aggravated in naïve infections and may be reduced in subsequent outbreaks; 2) following the index wave of outbreaks, subsequent challenges may be accompanied by modified human behaviors in which the affected farmers may not reports deaths; 3) early interventions in subsequent outbreaks may reduce the impacts observed compared to observations in previous outbreaks. 

Based on the seasonal calendars, the most common pig diseases occurred all the year round but ASF occurred cyclically from November to March. The months of October to November and January to February were recorded as months with lowest sale of pig and pork; in these periods, the stakeholders are primarily economizing for purposes of cropping in preparation for school fees and other school-related expenses in the following January. December may receive a surge in sale due to festivities (Christmas and New Year) and July to October were reported as months when supply of pigs from farmers are higher, because farmers were trying to earn in preparation for the October and November expenditures on cropping while at the same time minimizing on the risk of losing mature pigs in later parts of the year, the high-risk period for ASF. 

In the identified but complex calendars (Figure 3 and Figure 4), during the massive sale off of pigs due to fear of ASF, traders and middlemen took undue advantage of oversupplies and crash pig/pork prices; farmers, who were in dire needs of money, depended on income from livestock primarily since harvest period for crops are not yet due at this period. In the succeeding months (April–July/August), the survivor/retained piglets and weaners from the previous year are ready for the marketing and high prices supervened due to the scarcity of pigs, based on the demand–supply curve. The ensuing huge traffic of pigs to the livestock markets and abattoir provided ample room for unidirectional movements of pig-associated infections including ASF. It should be understood that the unsold pigs during the high supply period of July–Oct may return to the farms of origin after having had intense interactions and shared environment with pigs from multiple sources, a predisposition to disease outbreaks. In this case, the participatory epidemiology process has become a useful tool in troubleshooting the previously less understood concepts, facilitators and drivers supporting the perpetuation of the ASF virus in the Southern Highlands.

The slaughter slabs were particularly highlighted as high-risk points for disease introduction and transmission. Other workers have confirmed the major role of the slaughter slabs in the introduction and transmission of ASF [34,35,36]. In the current work, firstly, pigs arrived at the slaughter slabs routinely and, if unsold, were kept for a few days, closely interacting with other pigs, and were then returned to the originating farms. Secondly, the slabs were mostly upstream and contaminate water sources which serve the community downstream. A review of the past outbreaks had suggested evidence of the role of rivers and streams in ASF outbreaks, particularly in the Chunya and Songwe districts. We observed similar challenge in the current study in the visited slaughter slab and will suggest a thorough spatio-temporal analysis of the current situation. Avoidance of water contamination by wastewater from the abattoirs and slaughter slabs becomes critical to prevent the introduction of pathogens and reduce the extent of transmission during outbreaks of such diseases. Thirdly, the uncontrolled movement dynamics to and from the slaughter slabs were drivers of infections, since no stop-and-check practices or movement permit is carried out on such animals being transported. These identified hotspots for ASF infection and transmission can serve as critical control points for intervention to reduce the burden of ASF in the Southern Highlands. 

Anecdotally, outbreaks of ASF in neighboring countries of Malawi and sometimes Zambia are a risk factor for infection of border towns in the Southern Highlands in Tanzania. On at least three occasions, outbreaks in Karonga, Malawi, facilitated infections in Kyela and other inland districts of Tanzania (for example, Ileje, Mbeya, and Chunya). Similarly, outbreaks in Songwe and Tunduma have been linked with previous outbreaks of ASF in the Zambian borders. In view of these observations, the utilization of epi-zonal approach and joint border surveillance for selected high-impact diseases will be a good approach in combating the scourge of ASF in these joint border areas. 

The identified risk factors, drivers and facilitators of ASF infections have been listed in this work. The draft ‘*United Republic of Tanzania’s National African swine fever Strategy and Control Plan*, 2019’ already factored in several of the identified risk factors and drivers of infections. Its implementation should therefore benefit the country in its effort at controlling ASF in Tanzania, with benefits for rural livelihoods, food security and household incomes. The national and subnational governments should join forces to combat the endemicity of ASF virus in the Southern Highlands of Tanzania. However, because the government resources are always spread thin in developing economies like Tanzania, international supports from donor partners and efforts from the regional and continental bodies will be necessary to support the government in its effort to combat the unrelenting incidences of ASF in Tanzania. Finally, the stakeholders appreciated the engagement and ensued discussions and requested feedback and follow-up training programs on best practices on livestock husbandry and diseases control/biosecurity.

## 5. Conclusions

The Southern Highlands of Tanzania remains a strategic hub for pig production in East Africa but the cyclical outbreaks of ASF in this region continue to militate against progress in the sector, especially with the prevailing situation of biosecurity, trade dynamics and management practices. Using the identified issues in this paper, the national (including the subnational authorities), regional and international bodies should join efforts to support the control and eradication of ASF in Tanzania, particularly in the Southern Highlands. Such efforts should involve the policy makers, financiers and resource allocators, stakeholders in the industry, the veterinary authorities and disaster management offices. The issue of ASF should not be looked at from the prism of veterinary services alone, considering its implications on livelihoods, food and nutritional security, rural economies and disruption to regional trades and transboundary potential. We advocate the utilization of a participatory epidemiology and participatory disease search in such control and eradication efforts.

## Figures and Tables

**Figure 1 pathogens-09-00155-f001:**
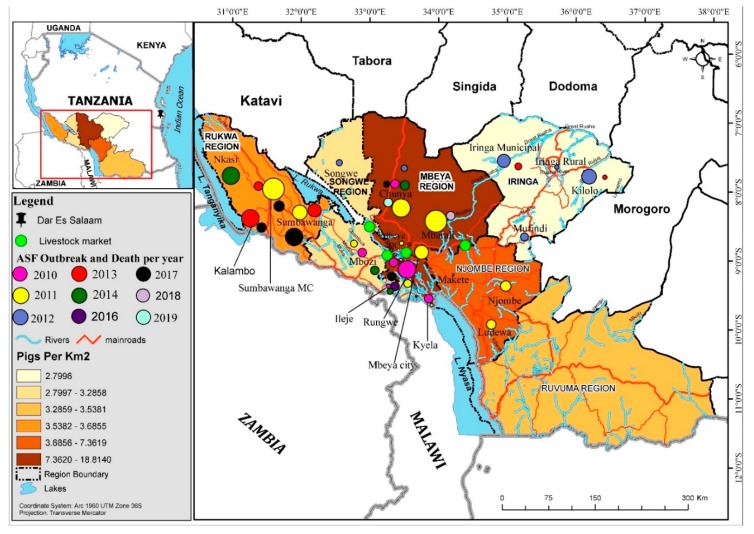
Spatial map of the Southern Highlands with significant pig-related and ASF locations, trade networks, markets, and high-density pig areas. Note that the size of the circle indicates the relative size of the outbreak impact by reported mortalities (see Figure 2a,b for numbers).

**Figure 2 pathogens-09-00155-f002:**
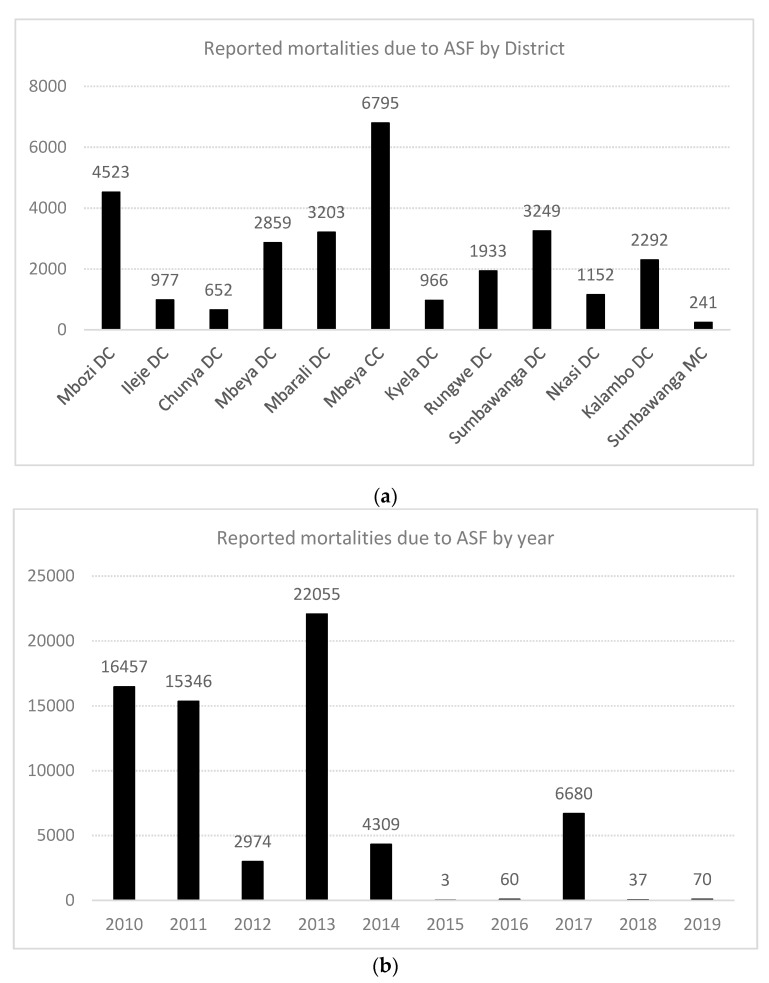
(**a**) Graphical representation of outbreaks of African swine fever (ASF) in the Southern Highlands, Tanzania, by numbers of reported pig deaths; (**b**) Reported pig deaths directly attributable to ASF outbreaks in Tanzania by year, 2010–2019. Note: DC = district council; MC = municipal council.

**Figure 3 pathogens-09-00155-f003:**
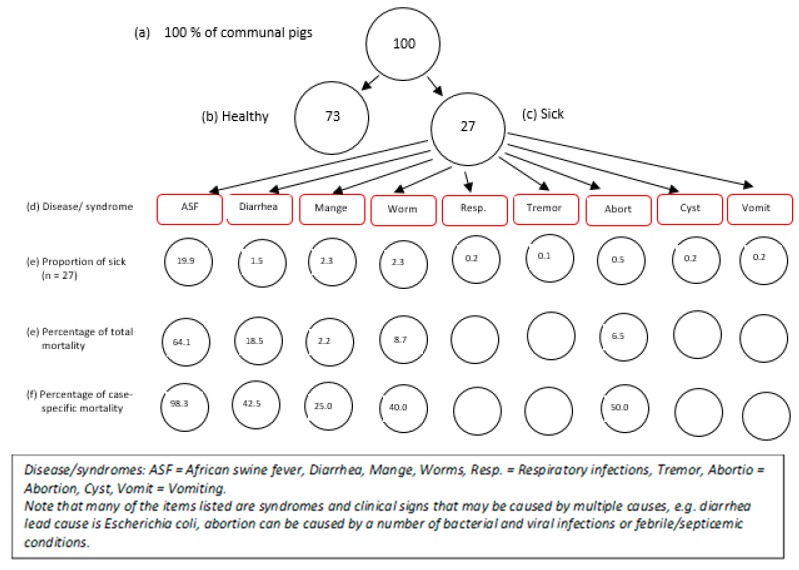
Proportional piles for morbidities and mortalities associated with common pig diseases, Southern Highlands, Tanzania.

**Figure 4 pathogens-09-00155-f004:**
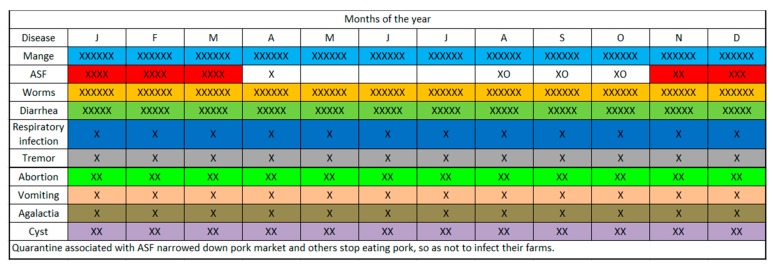
Seasonal calendar on occurrence of common pig diseases based on the knowledge and perceptions of pig stakeholders, Southern Highlands, Tanzania. The ‘xxx’ are indicators of consensus. The more ‘x’, the more the communities that commented on the disease in question. Some of the communities have not experienced some of the listed diseases and will not comment on them. Where ‘o’ was indicated, it meant that the community commented on a disease but did not have any experience with it. J, F, M, A…D are the months of the year.

**Figure 5 pathogens-09-00155-f005:**
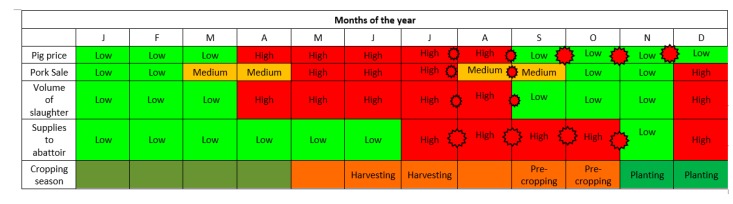
Seasonal calendar for determinants of pig sales and movements based on the knowledge and perceptions of pig stakeholders, Southern Highlands, Tanzania. 
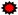
 The red stars on the timeline are indicative of a likely hotspot period, when the risk of ASF virus infections may escalate. The size of the star is indicative of the size of the risk; greater risk is indicated by bigger stars.

**Figure 6 pathogens-09-00155-f006:**
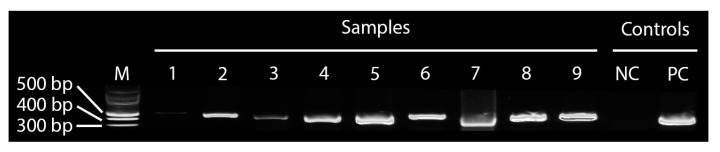
The PCR product on the agarose gel. The numbers on the lanes in the gel documentation (no. 1–9) correspond to (1) fetal fluid, (2) mesenteric lymph node, (3) abdominal fluid, (4) spleen 2, (5) spleen 1, (6) uterine fluid, (7) hepatic lymph node, (8) serum, and (9) whole blood. M is a DNA marker. NC is the negative control for master mix and PC is the ASF-positive control.

**Table 1 pathogens-09-00155-t001:** Quantitated proportional piles for livestock populations and economic contributions to rural livelihoods, Southern Highlands, Tanzania.

Animal Species Population Per 100 Livestock													
Diseases/Syndrome	Chickens	Cattle	Pigs	Ducks	Goats	Cats	Dogs	Rabbit	Sheep	Pigeon/Dove	G. fowl	Donkeys	G. pig
Mean ± SD	29 ± 20	21 ± 8	14 ± 6	4 ± 2	14 ± 5	4 ± 2	5 ± 3	2 ± 1	4 ± 2	5 ± 2	2 ± 1	3 ± 1	1 ± 0
95% CI	(10 to 48)	(14 to 29)	(9 to 19)	(2 to 5)	(9 to 18)	(2 to 6)	(2 to 8)	(1 to 3)	(2 to 5)	(2 to 8)	(1 to 3)	(−4 to 9)	(1 to 1)
Economic contributions of livestock species to household incomes and livelihoods by percentages *													
Mean ± SD	21 ± 11	19 ± 8	40 ± 18	3 ± 1	8 ± 2	3 ± 1	3 ± 1	2 ± 1	4 ± 3	4 ± 1	1 (NA)	3 ± 0	1 ± 1
95% CI	(10 to 33)	(11 to 27)	(21 to 58)	(1 to 4)	(5 to 10)	(1 to 5)	(2 to 4)	(−0.1 to 3)	(−3 to 12)	(2 to 5)		(3 to 3)	(1 to 1)

* Note that the contributions here are attributable to livestock species alone. Other sources of income including crop farming and public/private services are not included in the analysis. SD = standard deviation; CI = confidence interval.

**Table 2 pathogens-09-00155-t002:** Quantitated proportional piles for ranking of pig diseases based on impacts and economic implications, Southern Highlands, Tanzania.

	Common Pig Diseases (Importance Based on Ranking)/100 Pigs									
Diseases/Syndrome	ASF	Diarrhea	Mange	Worms	Respiratory Infection	Tremor	Abortion	Agalactia	Cyst	Vomiting
Mean ± SD	69.2 ± 12.5	6.8 ± 3.4	10.4 ± 9.3	9.6 ± 5.7	0.6 ± 0.9	0.2 ± 0.5	1.8 ± 4.0	0	0.8 ± 1.1	0.6 ± 1.3
95% CI	(53.7 to 84.7)	(2.6 to 11.1)	(−1.1 to 21.9)	(2.5 to 16.7)	(−0.5 to 1.7)	(−0.4 to 0.8)	(−3.2 to 6.8)	NA	(−0.6 to 2.2)	(−1.1 to 2.3)
	Common pig diseases (importance based on impact of morbidity and mortality)/100 pigs									
Mean ± SD	73.6 ± 14.7	5.4 ± 4.2	8.4 ± 7.6	8.6 ± 6.0	0.6 ± 0.9	0.2 ± 0.5	1.8 ± 4.0	0	0.8 ± 1.1	0.6 ± 1.3
95% CI	(55.3 to 91.9)	(0.2 to 10.6)	(−1.1 to 17.9)	(1.1 to 16.1)	(−0.5 to 1.7)	(−0.4 to 1)	(−3.2 to 6.8)	NA	(−0.6 to 2.2)	(−1.1 to 2.3)
	Common pig diseases (Importance based on frequency of occurrence)/100 pigs									
Mean ± SD	1.7 ± 2.1	5.2 ± 4.4	25.2 ± 24.9	62.7 ± 29.7	0.3 ± 0.5	0.3 ± 0.8	2.7 ± 6.1	0.5 ± 1.2	0.7 ± 1.2	0.8 ± 2.0
95% CI	(−0.5 to 3.8)	(0.6 to 9.8)	(−1.0 to 51.3)	(31.5 to 93.9)	(−0.2 to 0.9)	(−0.5 to 1.2)	(−3.7 to 9.0)	(−0.8 to 1.8)	(−0.6 to 1.9)	(−1.3 to 3.0)

Note: SD = standard deviation; CI = confidence interval.

**Table 3 pathogens-09-00155-t003:** Response to biosecurity questions based on in-depth interviews, Southern Highlands, Tanzania.

S. No.	Biosecurity Variable	Yes	No	Percentage Compliance (%)
1.	Notified neighbors during ASF outbreaks	25	12	68
2.	Notified veterinary and livestock field officers during ASF outbreaks	30	7	81
3.	Restricted access to all visitors during outbreaks	24	13	65
4.	Conducted communal sale of pork and sharing with neighbor	35	2	95
5.	Buried intestinal content following slaughter	5	32	14
6.	Allowed scavenger access to pig farm or around the farm	37	0	100
7.	Used common water source for the pigs	37	0	100
8.	Gate at entrance and fence	1	36	3
9.	Foot dips for disinfection before the house	0	37	0
10.	Record keeping	5	32	13
11.	Routine (regular) cleaning	2	35	5
12.	Quarantine newly purchased pigs for at least 10 days	4	33	10
13.	Safe disposal of feces and dead pigs (away from other animals)	7	30	19
14.	Remove manure and litter routinely	30	7	81
15.	Hand sanitizer, gloves and washing	0	37	0
16.	Usage of Disinfectant after cleaning	0	37	0
17.	Regular cleaning and disinfection of feeders, drinkers and equipment	0	37	0
18.	All-in all-out production system	0	37	0
19.	Separate sick pigs	0	37	0
20.	Conduct movement from young to older pigs	0	37	0
21.	Change rubber boots/slippers *	18	18	50
22.	Change clothing when going in/out of pig pen *	2	35	6
23.	Lock for each pen	35	2	95
24.	Assess health status of pigs using professionals	34	3	92
25.	Do not mix different ages	31	6	84
26.	Do not mix different species	35	2	95

A total of 37 IDIs was obtained for this analysis. * One response missing (n = 36).

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
