# Peer review of "Drivers, Risk Factors and Dynamics of African Swine Fever Outbreaks, Southern Highlands, Tanzania"

_pathogens, 2020, doi:10.3390/pathogens9030155_

Round 1
Reviewer 1 Report
This is an extremely interesting overview of the ASF situation in this part of Tanzania. It is particularly relevant to current global outbreaks of this disease. These data and description of disease dynamics in this African country deserve to be published and an open access forum is ideal to ensure maximum accessibility to all global veterinary authorities, however, there are serious flaws in the presentation of the small amount of numerical data included in this paper which lead me to have significant doubts about accepting it. The qualitative sections are also lacking clarity as to whether the topics described are based on interview responses or the authors' personal experience/opinions. For these reasons, I am recommended rejecting the paper at this stage but would encourage the authors to resubmit once they have corrected the apparent errors.
I am providing my review below and hope that it assists the authors with their revision.
(NB. There are no line numbers in my version, and the page numbers do not start until page 10! For this reason I have just used the section headings.)
While the English is generally good, there are a large number of minor errors, probably caused by a lack of proof-reading, and generally a rush to submit. Examples include being unsure whether they are using US or UK English (e.g. diarrhea (USA) and then changing to diarrhoea (UK) in the next paragraph), “infusion” of expert opinions instead of “introduction”, “most ranked” instead of most common or frequent, often using plural where singular is necessary, simple spelling errors, etc. Please check and correct the English throughout the document.
Overall, there are too many abbreviations for the reader to remember! An example of this is CSS for clinical signs and symptoms, which appears to only be used once or twice. Ditto, PE/PDS. However, I could not find a definition of DC which is used repeatedly (perhaps district county or something similar?).
Materials and Methods:
Section 2.2. It would be useful to provide a very short description of “proportional piling” here as most veterinary readers will be unaware of this practice.
Section 2.4. Were samples taken from animals that had already died or were animals euthanised/slaughtered? What were the described signs and symptoms?
Section 2.5. Here it states percentages and standard deviations, the mean is not mentioned.
Results: This is the biggest problem area, particularly with respect to the tables and figures
The first paragraph of the results section is describing the Tanzanian situation and should be in the introduction in my opinion (or perhaps in the discussion). These are not data that you have collected yourselves. It would probably be sufficient to state “pig densities in the XXX regions ranged from 2 to 6 pigs per km2”, similarly, a range is probably sufficient for the total number of pigs in each region.
You state that women were actively encouraged to take part in the group sessions, I would like to know how many women were moderators and/or veterinary officers included in the project management on the ground? The proportions of male to female moderators etc. would be relevant here as woman are often more likely to voice their opinions to other women.
Figure 1. Provides a large amount of useful information and helps readers who are not familiar with Tanzanian geography. However, the outbreak data is confusing: different sized circles are used, which presumably have some meaning with respect to the size of an outbreak – I cannot find any legend which explains this.
3.2. A map is required to illustrate this section. I am not aware how far away Dar es Salaam is from the Southern Highlands, or where these other regions are located, so I cannot follow this entire page. I think there is probably too much detail in this section, especially as the remainder of the paper concentrates solely on the Southern Highlands region. Is the historical background really part of the results of this study? Shouldn’t this be in the introduction?
Figure 2. Part A: I’m sorry but something has gone wrong with the creation of this figure, the data in this pie chart are nonsensical, e.g. 2859.10%? The picture resolution is so low that I did not notice these strange percentages at first, so that also needs to be improved. Part B: I can’t understand why the authors chose to illustrate a timeline of disease outbreaks as a pie chart! Conventional scientific practice would be a line diagram or even a bar chart if the cases/deaths are not considered to be linked to each other. Also, Part A appears to refer to the Southern Highlands only, whereas Part B includes cases throughout Tanzania? If this is the case, then why are these two graphics together?
3.3. Presumably pigs and the sale of pigmeat is primarily in the hands of men, whereas chickens are seen as women’s work and presumably women’s (and therefore the family’s) income? If this is the case, it might be worth adding it to explain this comment.
While your later data confirm that stakeholders were indeed excellent at accurately recognising ASF, this section seems misleading as how can they reliably distinguish between a pig with ASF or a pig with vomiting, diarrhoea or abortion (perhaps due to sudden death of the ASF pig)? Similarly, how are pigs with cysts recognised by farmers or is this based on the post-mortem findings?
Table 1. There are hardly any numerical data in this paper. Virtually no statistical analysis has been done and yet there appear to be mistakes in the calculation of the means in this table! If such simple errors are present here, it leads me to wonder if other less obvious errors are present elsewhere. Also, it is not explained why number 4 is “slaughter” rather than a village (perhaps the public slaughter facility is located in one of the villages?). And in the economic section of the table “Village 1” seems to be have been included twice with exactly the same data.
Table 2. There are row headings missing here, although the means appear to be correct (but perhaps they aren’t means, there is no heading!) Are these results also per village? The data relating to tremors are provided to 1 decimal place (as they are so low), but all other data are provided with no decimals.
Figure 3. Please include the disease names instead of the Roman numerals, it seems unnecessarily complicated to have to search for them in the legend.
Figure 4. Cannot be read in this picture definition. I cannot zoom in enough to read it properly. XO is not explained in the text.
Table 3. Is the gel phoresis picture supposed to be part of Table 3? If not, it needs a legend/title itself. Again, it is not legible in the manuscript.
3.6. This qualitative part is interesting and provides background to the ASF situation in Tanzania. However, some parts, e.g. neighbours settle scores by throwing carcases over fences, seems anecdotal. Is this section based on actual issues discussed during the group discussions and interviews or is this just the authors’ opinion/knowledge? This is not clear at present.
3.6.3. It would probably be better to include all of these percentages in a table rather than just listing them all in the text.
3.6.4. This part seems to jump backwards and forwards from markets to farm gate buyers and back again. The details on dead/dying/sick animals still entering the human food chain is very interesting and relevant to overall food safety. Again, is this the authors’ opinion or based on the participants’ experiences? The final paragraph does seem to report what butchers and farmers say about certain topics, but the first two paragraphs are not clear where this information is coming from.
Author Response
We thank the reviewer for the detailed comments. We have taken time to ensure that all comments are addressed as best as we can in this round.
There are serious flaws in the presentation of the small amount of numerical data.
-The data are now presented more clearly and bolder for a better view. A reread through the document will confirm this.
The qualitative sections are lacking clarity.
-The basis for the qualitative data is now presented at the beginning of the section.
There are no line numbers in my version, and the page numbers do not start until page 10! For this reason, I have just used the section headings.
-Line numbers are now added. On page number, the authors did not want to change the template formating. The insertion of landscape in-between the portrait format changed page number style that came with the template. We hope that the editorial office will correct this if the paper is accepted.
While the English is generally good, there are a large number of minor errors, probably caused by a lack of proof-reading, and generally a rush to submit. Examples include being unsure whether they are using US or UK English (e.g. diarrhea (USA) and then changing to diarrhoea (UK) in the next paragraph), “infusion” of expert opinions instead of “introduction”, “most ranked” instead of most common or frequent, often using plural where singular is necessary, simple spelling errors, etc. Please check and correct the English throughout the document.
-We reread the document to correct English and got an opinion of an English lecturer. We used American English throughout the document. “infusion” was changed to “introduction” and “most ranked” changed to "most common". While we want to make rapid communication in view of the fact that ASF is the most important animal disease ravaging the world currently and that countries with similar animal health platforms will benefit from this report, there was actually no rush to submit and that was why we chose this journal to have thoroughness and value addition with maximum visibility.
Overall, there are too many abbreviations for the reader to remember! An example of this is CSS for clinical signs and symptoms, which appears to only be used once or twice. Ditto, PE/PDS. However, I could not find a definition of DC which is used repeatedly (perhaps district county or something similar?).
-Abbreviations were now kept to minimum essentials of five viz: FGD, IDI, PE/PDS, ASF and RT PCR. Meaning were inserted on page 1 of the document. All other abbreviations were converted to full text to prevent distractions.
Materials and Methods:
Section 2.2. It would be useful to provide a very short description of “proportional piling” here as most veterinary readers will be unaware of this practice.
-Brief on the proportional piling is now included (see 2.3)
Section 2.4. Were samples taken from animals that had already died or were animals euthanised/slaughtered? What were the described signs and symptoms?
-The mentioned signs and symptoms are now described. The clinically sick pig was exsanguinated. Some other samples were from dead pigs.
Section 2.5. Here it states percentages and standard deviations, the mean is not mentioned.
-We meant, 'mean percentages and standard deviations'.
Results: This is the biggest problem area, particularly with respect to the tables and figures
The first paragraph of the result section is describing the Tanzanian situation and should be in the introduction in my opinion (or perhaps in the discussion). These are not data that you have collected yourselves. It would probably be sufficient to state “pig densities in the XXX regions ranged from 2 to 6 pigs per km2”, similarly, a range is probably sufficient for the total number of pigs in each region.
-We originally included this portion here because it was the outcome of the review aspect of our evaluation. It should serve as a background to further detailed information to be described here. Specifically, it rather described the Southern Highlands situation in relation to the Tanzanian pig populations. Based on the suggestion of the reviewer, it is now moved to the discussion section as an opening statement.
You state that women were actively encouraged to take part in the group sessions, I would like to know how many women were moderators and/or veterinary officers included in the project management on the ground? The proportions of male to female moderators etc. would be relevant here as woman are often more likely to voice their opinions to other women.
-Specifically, in 36.4 % of the groups, women were the lead respondents and approximately 50% of the veterinary/livestock officers were female. Women dominated 54.5% of the groups by population and one group was almost exclusively women.
Figure 1. Provides a large amount of useful information and helps readers who are not familiar with Tanzanian geography. However, the outbreak data is confusing: different sized circles are used, which presumably have some meaning with respect to the size of an outbreak – I cannot find any legend which explains this.
- Thank you for the observation. Different sizes of the circle indicated the size of impacts by pig deaths. It is now added as a footnote, 'Note that the size of the circle relatively indicates the size of the outbreak impact by reported mortalities (see figure 2 a & b for numbers)'.
3.2. A map is required to illustrate this section. I am not aware how far away Dar es Salaam is from the Southern Highlands, or where these other regions are located, so I cannot follow this entire page. I think there is probably too much detail in this section, especially as the remainder of the paper concentrates solely on the Southern Highlands region. Is the historical background really part of the results of this study? Shouldn’t this be in the introduction?
-Currently, we did a small map of Tanzania by the left and did a bigger call out a map of the Southern Highlands, the place of focus. We believe that this should be okay as putting two full maps (including that of Tanzania showing the whole regions) will seem too much. The document already have 6 figures and 3 extra pictures in the supplementary data and we are of the opinion that a full map of Tanzania will not add much value but it can be produced on demand.
Figure 2. Part A: I’m sorry but something has gone wrong with the creation of this figure, the data in this pie chart are nonsensical, e.g. 2859.10%? The picture resolution is so low that I did not notice these strange percentages at first, so that also needs to be improved. Part B: I can’t understand why the authors chose to illustrate a timeline of disease outbreaks as a pie chart! Conventional scientific practice would be a line diagram or even a bar chart if the cases/deaths are not considered to be linked to each other. Also, Part A appears to refer to the Southern Highlands only, whereas Part B includes cases throughout Tanzania? If this is the case, then why are these two graphics together?
-We acknowledge the confusion that arose from the previous version. We used pie chart because we are comparing proportion totalling 100% and pie chart can represent that. We acknowledge the poor quality picture and have enhanced it and now used bar chart instead. The 2859.10% meant (n = 2859, and 10% of the total) but the % is now taken out and numbers retained in the graph for all cases. The two graphs are necessary as one represented the year and one the district.
3.3. Presumably pigs and the sale of pigmeat is primarily in the hands of men, whereas chickens are seen as women’s work and presumably women’s (and therefore the family’s) income? If this is the case, it might be worth adding it to explain this comment.
- The comment is very well taken and now added to the discussion. Thank you for your perspective.
While your later data confirm that stakeholders were indeed excellent at accurately recognising ASF, this section seems misleading as to how can they reliably distinguish between a pig with ASF or a pig with vomiting, diarrhoea or abortion (perhaps due to sudden death of the ASF pig)? Similarly, how are pigs with cysts recognised by farmers or is this based on the post-mortem findings?
-The stakeholders were only able to recognise disease through signs, symptoms and syndromes. They were also with the attending vets when slaughtered animals are rejected due to multiple cysts and can see these and relate to it due to losses they incurred. It will be good to note and document this because they can become the first line of passive surveillance for animal diseases. It is however acknowledged that they are not expert in clinical diagnosis.
Table 1. There are hardly any numerical data in this paper. Virtually no statistical analysis has been done and yet there appear to be mistakes in the calculation of the means in this table! If such simple errors are present here, it leads me to wonder if other less obvious errors are present elsewhere. Also, it is not explained why number 4 is “slaughter” rather than a village (perhaps the public slaughter facility is located in one of the villages?). And in the economic section of the table “Village 1” seems to be have been included twice with exactly the same data.
-The statement here is understandable but misconstrued. First, all statistics done were basically descriptive (mean, standard deviations, percentages and such) and they appear as best as can be done for limited data arising from group-oriented information sources. No error in the means and standard deviations. There is no need to introduce complex statistics if it will add no value and if it may produce skewed results. If we have enough individual verifiable details that justified such complex statistics, we may conduct analytical statistics. The error in the economic contribution is regretted. It was introduced through 'insertion of the next row' to create an opportunity for data entry but did not invalidate the results. We rechecked the result and it is still valid.
Table 2. There are row headings missing here, although the means appear to be correct (but perhaps they aren’t means, there is no heading!) Are these results also per village? The data relating to tremors are provided to 1 decimal place (as they are so low), but all other data are provided with no decimals.
- The heading was indicated and fell into the previous page earlier. It is adjusted to sit on top f the table now. The values were raw percentages by proportion from the group and the mean % + SD were calculated below and 95% Confidence intervals were inputted.
Figure 3. Please include the disease names instead of the Roman numerals, it seems unnecessarily complicated to have to search for them in the legend.
-The decision to put in roman numerals and list the disease below was because many of the diseases will not fit the box or if fitted in, will be too small to read. We have now made an effort to fit in the disease or syndrome names.
Figure 4. Cannot be read in this picture definition. I cannot zoom in enough to read it properly. XO is not explained in the text.
-The picture is made more definitive for ease of reading. The meaning of 'X' and 'O' are indicated in the footnote.
Table 3. Is the gel phoresis picture supposed to be part of Table 3? If not, it needs a legend/title itself. Again, it is not legible in the manuscript.
-The gel phoresis is now separated as a separate figure 6. It, however, speaks to Table 3. All pictures are now enhanced.
3.6. This qualitative part is interesting and provides background to the ASF situation in Tanzania. However, some parts, e.g. neighbours settle scores by throwing carcases over fences, seems anecdotal. Is this section based on actual issues discussed during the group discussions and interviews or is this just the authors’ opinion/knowledge? This is not clear at present.
- Thank you, we have now started the section on 'Qualitative evaluations with the statement, 'The information detailed below arose from the qualitative interviews conducted during the FGDs and the IDIs, backed up by key informants where detailed information was lacking. The details represent the opinions of the stakeholders and not tat of the interviewers or authors'. We actually needed a second opinion on this aspect of 'settling scores'. We wanted to capture opinion got from the people as best as not to lose anything but we also believe that the evidence is just anecdotal. We have deleted that statement.
3.6.3. It would probably be better to include all of these percentages in a table rather than just listing them all in the text.
- We have now introduced Table 4 that contains the summary of the percentages.
3.6.4. This part seems to jump backwards and forwards from markets to farm gate buyers and back again. The details on dead/dying/sick animals still entering the human food chain is very interesting and relevant to overall food safety. Again, is this the authors’ opinion or based on the participants’ experiences? The final paragraph does seem to report what butchers and farmers say about certain topics, but the first two paragraphs are not clear where this information is coming from.
- We thank the reviewer for the observation. In the pig value chain, there can be no dividing line between each operation based on what was reported by the stakeholders. They crisscrossed from farms to markets to slaughter facilities and back and we have done our best to streamline the reports. It follows Farm/farm areas and on farm sales to sale in outbreak periods to slaughter facilities to distribution channels and to homesteads. These narratives are the complete story for epidemiologists to make a complete link and we are doing a follow-up to do a comprehensive spatial-temporal analysis now.
Overall, we thank the reviewer for critical analysis of the document. His/her inputs have added value to the document and pointed out the earlier overlooked errors and mistakes that may have passed inadvertently.

Reviewer 2 Report
The manuscript “Drivers, risk factors, and dynamics of African swine fever outbreaks, Southern Highland, Tanzania” sent to me for revision is dedicated to the detailed identification of factors affecting recurrent of ASF epidemics in The Southern Highland zone of Tanzania, assessment of their impact on the situation of the local population, as well as to knowing of the state of knowledge among breeders, traders and processors about this disease and possible preventive measures.
African swine fever remains an important pig disease all over the world due to the speed of spreading, economic impacts and food implications. Tanzania region described in the manuscript can be classified as an important pig producing hub in East Africa. Repetitive ASF epidemics in this region have prompted the authors to take a closer look at the reasons for this.
A number of research results collected on the basis of methodically conducted interviews among persons interested and directly affected by this problem were collected and discussed in the paper. Based on a thorough analysis of the data obtained, a number of factors affecting cyclically recurring epidemics have been presented. After presenting a comprehensive discussion on the obtained information, the work ends with conclusions on the possible measures to be used to limit the ASF epidemic in this region. The authors point out the need to carry out a wide-ranging information campaign on the correct behavior to limit the occurrence and spread of ASF. They also point out that one should not limit oneself only to issues related to veterinary services, but one should look at the problem from a much broader perspective.
It is necessary to engage in solving the problem not only of local authorities but assistance at the state level is necessary, and given the economic status of Tanzania, the involvement of international institutions should also be expected.
From the point of view of a reader from a country where there is not such a big problem with ASF, the data presented on the awareness and behavior of people directly involved in the breeding, distribution and slaughter of pigs, whose economic status directly depends on the reduction of the ASF epidemic, are quite frightening.
Perhaps the information presented in the manuscript will prompt the relevant global authorities to become more interested in this problem and to intensify efforts to implement procedures to limit and control the spread of ASF in Tanzania.
The reported work is interesting for the scientific community and not only and the submitted version of the manuscript may be considered suitable for the Journal after revision, provided the authors pay due attention to improving the quality of Figures presented in the manuscript. In their present form, they are not very clear and difficult to read. Careful text editing is also recommended.
Author Response
We thank the reviewer for detailed comments on the manuscript and for considering the manuscript worthy of consideration for publication. Specifically, the reviewer had requested the following:
Improvement of the introduction to capture sufficient background and include all the relevant references. Clearer presentation of the results. Clearer/sharper images. Careful editing of the texts.
We have therefore done the following:
The introduction section has been subjected to a thorough review for its content. While we avoid being unnecessarily verbose and wordy, we have added relevant and current content. We believe that since this manuscript will be part of a special collection on ASF, other manuscripts will also detail the historical perspectives on ASF. We, therefore, are careful of unnecessary repetitions. Results have also been reviewed and made much simple and clearer to provide the succinct details. Clearer images are now provided where they were blurry or too small to read earlier. See figure 3 and the accompanying figure in Table 3. The whole of the write-up is now subjected to a reread by an English editor for grammar and tenses. We made an effort to remove unnecessary duplications, yet the essential details captured from the stakeholders were retained. It should benefit the pig industry globally.

Reviewer 3 Report
Excellent participatory epidemiology research paper in a setting with scarce official data. Very interesting, well described and scientifically sound. I would only improve the quality of the figures some parts of which are difficult to read at the moment (i.e. use white font for dark background). Revise also the numbering of pages. There is a typo in a "neiguhbours" in page "5 of 21".
Author Response
We are grateful to the reviewer for a good job.
The reviewer will want the following:
English language and style are fine/minor spell check required.
Improvement in the quality of the figures and readability if the unclear ones.
We have subjected the document to language revision by an English Editor.
We also improved the figures for those unclear ones (See Figure 3 and Figure in Table 3).
Reviewer 4 Report
In this manuscript by Fasina et al. , the authors set out to examine the factors underlying and facilitating the spread of African Swine Fever (ASF) in the Southern Highland (SH) region of Tanzania, based on “participatory epidemiology” and “participatory disease surveillance” (PE/PDS) involving veterinarians and stakeholders (farmer, butchers, traders..).
Main findings.
Outbreaks of ASF were identified from 2010 to 2019 and their geographical locations were mapped, allowing the authors to suggest some possible epidemiological links. Among pig diseases, ASF had the highest impact (morbidity/mortality and economic impact), while its frequency of occurrence was far lower than those of worm infections and mange. ASF occurred cyclically, prevailing from November to March.- Samples were collected in a farm were clinical symptoms of ASF were reported. All samples were PCR-positive for ASF. Quantitative evaluations were conducted in order to identify risk factors.
The manuscript contains interesting information, but it is very descriptive and much too long, with too much details and lacking a synthetic view.
Below are the specific remarks (difficult because of the lack of page numbering and line numbering)
Major remarks
Most paragraphs are very long. The information could be clearer if presented in tables, and not all the details are necessary.
3.2 Historical background. This paragraph is much too long, and difficult to follow for anybody not familiar with the map of SH, Tanzania. Perhaps it would be possible to combine geographical and historical data on a geographical map.
3.6.1. The second paragraph mostly deals with biosecurity measures (or lack of biosecurity measures), which are the subject of 3.6.3.
The discussion is much too long.
Author Response
We also appreciate the comments of reviewer 3. Specifically, he/she wants the following:
- English language and style are fine/minor spell check required
Clear presentation of results The conclusion must be supported by the results.
The manuscript and its paragraphs are too long and lack a synthesis view. Not all the details are necessary. Historical background and discussion are too long. Combine geographical and historical background.
We have done the following:
Similar to the responses to the two other reviewers, we reviewed the English and edited it. We have merged the Spatio-temporal portion with historical perspectives as suggested. Where necessary, we cut down significantly on the discussion and results, however, we are careful not to lose the essence provided by the stakeholders and captured in the result.
Round 2
Reviewer 1 Report
I would like to thank the authors for responding to the majority of my comments and greatly improving their manuscript. The data in the tables have now been corrected and their presentation in the text and figures is also much better. The English still needs a thorough proof-read, however, as there are many small errors (particularly in plural use as previously stated). In order to help the authors, I have provided some English corrections below, as well as a small number of content suggestions.
(Please note: I have not and cannot correct the entire document, it still needs proof-reading and correcting throughout. Once proof reading has been done, the person responsible would normally be mentioned in the Acknowledgement section.)
Page numbers can be included on all pages by adding section breaks (rather than page breaks) before and after the horizontal pages.
Introduction
Line 60: delete “the” before ASF
Line 78: delete especially, it would be better to write “…in 1987 and 1988, in particular, in the regions of XXX….”
Line 83: Write “detailed” instead of details
Line 85: Delete “as”, just “from” is enough.
Line 98-99: Correct to “ramifications” and “livestock systems”
M&M
Line 113: Correct to “…village must have been located…”
Line 125: Add i.e. before “no new issue…”
Line 133: Correct to “Narrative questions…”
Line 134: Correct to “…who administered the questionnaire were trained in the use…”
Line 151: Add “(for details, see section 2.3 below)” after proportional piling
Line 157: Delete described
Line 169: Correct spelling to “wards affected”
Line 177: “Off feed” would be “anorexia” in veterinary terminology, correct this here
Line 179: Additional information on the other tissue sources is required here, there are only 9 samples so it should be simple to add how many were from sick or deceased animals. I assume only the one pig was pregnant? Otherwise this should also be mentioned.
Line 192: Should this be “from proportional piles”? Correct spelling.
Results
This section is greatly improved and easier to understand.
Line 206: correct to “an ongoing outbreak”
Figure 1: While I agree with the authors that a detailed map would take up too much space, I would recommend adding Dar es Salaam to the small map of Tanzania on the left to give readers some perspective as to how far away this city is from the Southern Highlands.
Line 219: Correct was to “outbreaks of ASF were”
Section 3.1. Re. this whole historical section – please see my previous comment – is it really necessary to go into this much detail? Almost 1.5 pages? Especially the parts which are not relevant to the Southern Highlands? It seems much too long, please consider shortening this section. I think other reviewers also requested this.
Figure 2: This is greatly improved. Please correct the legend to “by number of reported pig deaths” as you no longer show the percentages.
Line 293: Guinea pigs and guinea fowl do not require capital “G”.
Line 296: As previously requested, please correct least ranked to “least common”
Line 301: correct to “…fast cash earner…”
Table 1: Thank you for removing the mistakes from this table. The means now appear to be correct. To clarify my previous comment: I was not requesting that the authors carry out additional statistical testing which would be meaningless in such a study, but was merely stating that if there are very few numerical data in a manuscript and very simple descriptive statistics (such as % and mean), then I would expect them to be correct and correctly presented in figures and tables!
Table 2: As previously requested, please present the means and 95% CIs to 1 decimal place to be consistent. The horizontal row headers are still missing. The authors’ responses state that this is per group but there are 5 groups in this table compared to 6 villages so I am still unsure where these numbers came from. Correct the legend as follows “…95% confidence intervals in parentheses”
Line 327: Replace quantitation with “quantitative analysis”
Figure 3: The addition of the disease names makes the figure much easier to read. The resolution is not as good as the other figures though, it is still quite blurry. In my version, there is a vertical line in the 100 circle which needs to be deleted. I would suggest “Abort.” as an abbreviation for abortion.
Figure 4: The resolution of this and Figure 5 is much improved, now I can finally read them. Diarrhoea is still written in UK English. Change the legend to “…others stop eating pork, so as not to infect their farms”
Figure 5: Do the different sizes of the stars have any specific meaning? A larger star means higher risk, perhaps? If not, make them all a uniform size or add some information to the legend.
Line 354: Correct to “…indicating that the farmers’….”
Line 355: I would not use the word perfect. I agree that farmers and other stakeholders appear to have a very good knowledge of ASF symptoms, but you have only tested 9 tissue samples (from how many pigs?) so I think “perfect” is an exaggeration. I would suggest changing this to “extremely good”.
Line 356: Correct to “…ASF, indicating transplacental transmission.”
Line 365-367: Separate the legend text. Move the legend relevant to Table 3 directly under this table (delete “in the Table 3 above”) and keep the legend relevant to Figure 6 here. Also correct the text as follows “…correspond to the serial number in Table 3 above.”
Line 373: Who were the key informants? Veterinary officers?
Line 374: Correct to “stakeholders and not those of the interviewers or authors.”
Line 376: Correct to “…farmers keep mixed breeds…”
Line 393: Correct to “….the farmers’ motivation to report outbreaks is low.”
Line 394: Correct to “…farmers claimed to implement movement restrictions…”
Line 399 and throughout the document: Correct visceral to “viscera”
Line 415: Correct to “…similar to those in most infected communities.”
Section 3.5.3., line 421-436: My previous suggestion to include a table here was so that you could reduce some of the text in this section. The table is an excellent addition which gives readers a helpful overview of the lack of biosecurity. However, it is not necessary to repeat all the data from the table in the text. Please choose the most important results to include in the text here, perhaps 3 or 4 topics? (Please do not start sentences with numbers).
Line 447: Delete” (7th and 24th of each month)” as this is irrelevant.
Entire page (and entire manuscript): check throughout for confusion between plurals and singular
Line 499: Change stemmed to “prevented”
Discussion
Line 521: Correct to “…present work has shed light…”
Line 534: I’m not sure what the authors mean by “the foregoing”, suggest changing this to “In view of these factors…”
Line 538: While I’m pleased that the authors agree with my statement regarding chickens belonging to women and pigs to men, I am happy for the authors to alter my text to fit the actual situation in Tanzania if necessary.
Line 552: Correct to “…Highlands were poor…” and delete “right from the farm to fork”
Line 561: Correct to “…were positive for ASF using PCR.”
Line 577: Delete “…of the following year”
Line 602: Correct to “…a few days, closely interacting with other pigs, and were then returned to the…”
Author Response
We thank the reviewer once again for the dedication and thoroughness to review the manuscript again. We have now responded to the extra comments as follows:
Introduction
Line 60: delete “the” before ASF
Response: ‘the’ now deleted.
Line 78: delete especially, it would be better to write “…in 1987 and 1988, in particular, in the regions of XXX….”
Response: ‘especially’ now deleted. It now reads, ‘Major epidemics of ASF were reported, in 1987 and 1988 in the regions of Mbeya and Arusha/Kilimanjaro……’
Line 83: Write “detailed” instead of details
Response: ‘details changed to detailed’.
Line 85: Delete “as”, just “from” is enough.
Response: ‘as’ now deleted.
Line 98-99: Correct to “ramifications” and “livestock systems”
Response: Corrected as suggested.
M&M
Line 113: Correct to “…village must have been located…”
Response: Corrected as suggested.
Line 125: Add i.e. before “no new issue…”
Response: ‘i.e.’ now added before ‘no new issue’.
Line 133: Correct to “Narrative questions…”
Response: Corrected as suggested.
Line 134: Correct to “…who administered the questionnaire were trained in the use…”
Response: Corrected as suggested.
Line 151: Add “(for details, see section 2.3 below)” after proportional piling
Response: Added as suggested.
Line 157: Delete described
Response: ‘described’ now deleted.
Line 169: Correct spelling to “wards affected”
Response: Corrected as suggested.
Line 177: “Off feed” would be “anorexia” in veterinary terminology, correct this here
Response: ‘Off feed’ changed to ‘anorexia’.
Line 179: Additional information on the other tissue sources is required here, there are only 9 samples so it should be simple to add how many were from sick or deceased animals. I assume only the one pig was pregnant? Otherwise this should also be mentioned.
Response: Additional information now provided to read, ‘A clinically sick but pregnant pig was purchased and euthanised by exsanguination. Spleen, mesenteric and gastro-hepatic lymph nodes, visceral fluid, fetal fluids, uterine fluids, spleen, serum and whole blood from the euthanized pregnant pig; and spleen from a diseased pig during the previous outbreak were collected for ASF confirmation using molecular methods.’
Line 192: Should this be “from proportional piles”? Correct spelling.
Response: “form the proportional piles” changed to “from the proportional piles”.
Results
This section is greatly improved and easier to understand.
Thank you for the encouraging comments and value addition to the work.
Line 206: correct to “an ongoing outbreak”
Response: ‘On-going’ changed to ‘ongoing’.
Figure 1: While I agree with the authors that a detailed map would take up too much space, I would recommend adding Dar es Salaam to the small map of Tanzania on the left to give readers some perspective as to how far away this city is from the Southern Highlands.
Response: A correction for the map has been requested and the revised map will be entered into the finalized reviewed document.
Line 219: Correct was to “outbreaks of ASF were”
Response: Corrected as suggested.
Section 3.1. Re. this whole historical section – please see my previous comment – is it really necessary to go into this much detail? Almost 1.5 pages? Especially the parts which are not relevant to the Southern Highlands? It seems much too long, please consider shortening this section. I think other reviewers also requested this.
Response: Corrected as suggested. Section is significantly reduced with a supplementary note for whosoever is interested in the details. We believe that the detail may be important for future works especially for spatio-temporal epidemiologists.
Figure 2: This is greatly improved. Please correct the legend to “by number of reported pig deaths” as you no longer show the percentages.
Response: Corrected as suggested.
Line 293: Guinea pigs and guinea fowl do not require capital “G”.
Response: Corrected as suggested. All ‘G’ changed to ‘g’ except where it begins a sentence.
Line 296: As previously requested, please correct least ranked to “least common”
Response: Corrected as suggested.
Line 301: correct to “…fast cash earner…”
Response: Corrected as suggested.
Table 1: Thank you for removing the mistakes from this table. The means now appear to be correct. To clarify my previous comment: I was not requesting that the authors carry out additional statistical testing which would be meaningless in such a study, but was merely stating that if there are very few numerical data in a manuscript and very simple descriptive statistics (such as % and mean), then I would expect them to be correct and correctly presented in figures and tables!
Response: Thanks for your comments. Based on the other reviewer’s comments too, we have now reduced the tables 1 and 2 to the essential details alone.
Table 2: As previously requested, please present the means and 95% CIs to 1 decimal place to be consistent. The horizontal row headers are still missing. The authors’ responses state that this is per group but there are 5 groups in this table compared to 6 villages so I am still unsure where these numbers came from. Correct the legend as follows “…95% confidence intervals in parentheses”
Responses: See the immediate response above.
Line 327: Replace quantitation with “quantitative analysis”
Response: Corrected as suggested
Figure 3: The addition of the disease names makes the figure much easier to read. The resolution is not as good as the other figures though, it is still quite blurry. In my version, there is a vertical line in the 100 circle which needs to be deleted. I would suggest “Abort.” as an abbreviation for abortion.
Response: Corrected as suggested.
Figure 4: The resolution of this and Figure 5 is much improved, now I can finally read them. Diarrhoea is still written in UK English. Change the legend to “…others stop eating pork, so as not to infect their farms”
Response: ‘Diarrhoea’ is changed to ‘diarrhea’. Legend also changed as suggested.
Figure 5: Do the different sizes of the stars have any specific meaning? A larger star means higher risk, perhaps? If not, make them all a uniform size or add some information to the legend.
Response: This is now added to read, ‘The size of the star is indicative of size of risk, higher risks for bigger stars.’
Line 354: Correct to “…indicating that the farmers’….”
Response: Corrected as suggested.
Line 355: I would not use the word perfect. I agree that farmers and other stakeholders appear to have a very good knowledge of ASF symptoms, but you have only tested 9 tissue samples (from how many pigs?) so I think “perfect” is an exaggeration. I would suggest changing this to “extremely good”.
Response: Corrected as suggested.
Line 356: Correct to “…ASF, indicating transplacental transmission.”
Response: Corrected as suggested.
Line 365-367: Separate the legend text. Move the legend relevant to Table 3 directly under this table (delete “in the Table 3 above”) and keep the legend relevant to Figure 6 here. Also correct the text as follows “…correspond to the serial number in Table 3 above.”
Response: This is done as suggested but also harmonized with the other reviewer’s comment who suggested the removal of the table with a note to indicate the table’s content.
Line 373: Who were the key informants? Veterinary officers?
Response: The key informants were veterinary and livestock field officers. This is reflected in the document now.
Line 374: Correct to “stakeholders and not those of the interviewers or authors.”
Response: Corrected as suggested.
Line 376: Correct to “…farmers keep mixed breeds…”
Response: Corrected as suggested.
Line 393: Correct to “….the farmers’ motivation to report outbreaks is low.”
Response: Corrected as suggested.
Line 394: Correct to “…farmers claimed to implement movement restrictions…”
Response: Corrected as suggested.
Line 399 and throughout the document: Correct visceral to “viscera”
Response: Corrected as suggested.
Line 415: Correct to “…similar to those in most infected communities.”
Response: Corrected as suggested.
Section 3.5.3., line 421-436: My previous suggestion to include a table here was so that you could reduce some of the text in this section. The table is an excellent addition which gives readers a helpful overview of the lack of biosecurity. However, it is not necessary to repeat all the data from the table in the text. Please choose the most important results to include in the text here, perhaps 3 or 4 topics? (Please do not start sentences with numbers).
Response: Section is now significantly shortened with reference to the table.
Line 447: Delete” (7th and 24th of each month)” as this is irrelevant.
Response: Deleted as suggested.
Entire page (and entire manuscript): check throughout for confusion between plurals and singular
Response: We have made effort to read and correct grammar and tenses confusion again but will take further suggestions from the editorial board and reviewers.
Line 499: Change stemmed to “prevented”
Response: Changed as suggested.
Discussion
Line 521: Correct to “…present work has shed light…”
Response: Corrected as suggested.
Line 534: I’m not sure what the authors mean by “the foregoing”, suggest changing this to “In view of these factors…”
Response: Changed as suggested.
Line 538: While I’m pleased that the authors agree with my statement regarding chickens belonging to women and pigs to men, I am happy for the authors to alter my text to fit the actual situation in Tanzania if necessary.
Response: We retained the statement with a slight modification to read ‘Presumably, pigs and the sale of pork are primarily the responsibilities of men, whereas chickens are seen as women’s works and a source of family income. This might have influenced the decisions of some of the women who insisted that chicken possibly play more role than pigs as sources of family income’.
Line 552: Correct to “…Highlands were poor…” and delete “right from the farm to fork”
Response: Corrected as suggested.
Line 561: Correct to “…were positive for ASF using PCR.”
Response: Corrected as suggested.
Line 577: Delete “…of the following year”
Response: Deleted as suggested.
Line 602: Correct to “…a few days, closely interacting with other pigs, and were then returned to the…”
Response: Corrected as suggested.

Reviewer 4 Report
The manuscript is still much too long, with too much details, and the authors should make every effort to shorten it by about 50%.
While it contains interesting information, it lacks a synthetic view and there are several redundancies, repetitions.
More specifically:
Historical perspectives (lines 212-280 and Fig. 2). Each paragraph is dedicated to a specific district or region that the reader cannot know if not familiar with the detailed geography of the area and its villages (lines 221-22, 236, 246-47, 257, etc). A synthetic view is lacking. The authors should try to describe the dynamics of the epidemics that occurred in 2010-2011, 2013-2014, 2017 and 2019, with perhaps an individual map for each of these waves.
The map in Fig 1 is very difficult to read, and Fig 2a brings no useful information.
Table 1 contains lots of unnecessary details. It could be replaced with a table of two lines: mean +/- SD and 95%CI for animal species, and the same for economic contributions.
Table 2 also contains unnecessary details. It could be replaced with a table of three lines: mean +/- SD and 95%CI for common pig diseases ranking / morbidity and mortality / frequency of occurrence).
Table 3 could be replaced by one or two sentences: “All samples were found positive by PCR amplification of the VP72 gene, with the highest positivities (lowest Ct values, close to 20), recorded for spleen whole blood and serum.”
3.5.1 Most of this paragraph (lines 376-405) in fact deals with biosecurity practices (or lack of biosecurity practices). Paragraphs 3.5.1, 3.5.2 and 3.5.3 should be combined in one or two paragraphs “Biosecurity practices at the slaughtering / Biosecurity practices at the farm level (including detection and notification of cases, restrictions, disposal of waste and manure …etc).
3.5.4; This paragraph could be more synthetic, with a clearer division into (i) movements of live pigs towards markets sites and around these sites, (ii) movements of live pigs towards slaughtering sites and around these sites (iii) movements of pig products from the slaughtering sites (including wastewates) (iv) movements of persons into the slaughtering sites and from the slaughtering sites, (v) movements of materials (knives, trucks).
The discussion is still much too long. The authors should make every effort to shorten it and to give a more synthetic view.
Author Response
Dear Reviewer,
We thank you for your insight and critical analysis. We have made an effort to comply with the requests made as follows:
Q1: Historical perspectives (lines 212-280 and Fig. 2). Each paragraph is dedicated to a specific district or region that the reader cannot know if not familiar with the detailed geography of the area and its villages (lines 221-22, 236, 246-47, 257, etc). A synthetic view is lacking. The authors should try to describe the dynamics of the epidemics that occurred in 2010-2011, 2013-2014, 2017 and 2019, with perhaps an individual map for each of these waves.
Response: We have now reordered the section based on your comment and that of the other reviewer. We summarised the details in the section to just 346 words now and provided details as supplementary notes for whoever want to read the details. We believe that this will shorten this section while preserving valuable dataset which may provide good historical background in future studies.
Q2: The map in Fig 1 is very difficult to read, and Fig 2a brings no useful information.
Response: The map (Fig 1) is reflective of the historical outbreaks 2011-2019 in the Southern Highlands and incorporated the important watercourses, livestock markets, and also shows outbreaks per year and magnitude. We believe that this should be of interest to an average epidemiologist who may want to follow in details later. Doing the map per year will seem superfluous as it will extend the figures in the paper to a total of 9 or 10 and you will need to relate each map to another to get spatial relativities of outbreaks. We believed that Figures 2a&b are also useful and the other reviewers seemed to concur with this assertion. Carefully check again.
Q3: Table 1 contains lots of unnecessary details. It could be replaced with a table of two lines: mean +/- SD and 95%CI for animal species, and the same for economic contributions.
Response: We complied with this instruction. Thank you for your perspectives. We originally keep the details to inform training and repeatability.
Q4: Table 2 also contains unnecessary details. It could be replaced with a table of three lines: mean +/- SD and 95%CI for common pig diseases ranking / morbidity and mortality / frequency of occurrence).
Response: We complied with this instruction. Thank you for your perspectives.
Q5: Table 3 could be replaced by one or two sentences: “All samples were found positive by PCR amplification of the VP72 gene, with the highest positivities (lowest Ct values, close to 20), recorded for spleen whole blood and serum.”
Response: We complied with this instruction. Thank you for your perspectives. We also provided the necessary details supporting figure 6 now which was lost because the table was deleted.
Q6: 3.5.1 Most of this paragraph (lines 376-405) in fact deals with biosecurity practices (or lack of biosecurity practices). Paragraphs 3.5.1, 3.5.2 and 3.5.3 should be combined in one or two paragraphs “Biosecurity practices at the slaughtering / Biosecurity practices at the farm level (including detection and notification of cases, restrictions, disposal of waste and manure …etc).
Response: This section is now reorganised under one subtitle as suggested. It was also reduced significantly to only 4 essential paragraphs
Q7: 3.5.4; This paragraph could be more synthetic, with a clearer division into (i) movements of live pigs towards markets sites and around these sites, (ii) movements of live pigs towards slaughtering sites and around these sites (iii) movements of pig products from the slaughtering sites (including wastewates) (iv) movements of persons into the slaughtering sites and from the slaughtering sites, (v) movements of materials (knives, trucks).
Response: We complied and reduced the section as suggested with a clear division into subsection as indicated. Thank you very much for adding value to the manuscript.
Q8: The discussion is still much too long. The authors should make every effort to shorten it and to give a more synthetic view.
Response: Effort is now made and the discussion is reduced where necessary. It should be known that this discussion is perhaps the most critical element in this work in order to engender biosecurity and risk mitigation messages to stakeholders. We will not want to lose any important feature of the discussions.

Round 3
Reviewer 4 Report
The authors have modified their manuscript according to the suggestions, and I particularly appreciate the clear division of paragraph 3.5.2 in subsections.
Some minor remarks :
Lines 383-84. The sentence is grammatically incorrect.
Lines 412-415 of paragraph 3.5.2(iii) mostly repeat the preceding paragraph, while this (iii) paragraph should deal instead with movement of pig products (meat, intestine, trotters and also wastewater).
Lines 428-432 should also belong to the previous paragraph (iii).
Author Response
Dear Editor,
We thank the reviewer for rapid action on the document.
We have now addressed the comments as follows:
Q1: Lines 383-84. The sentence is grammatically incorrect.
Response: Revised to read, 'While some districts have resident livestock markets located within them, some other districts share livestock markets with neighboring districts.'
Q: Lines 412-415 of paragraph 3.5.2(iii) mostly repeat the preceding paragraph, while this (iii) paragraph should deal instead with movement of pig products (meat, intestine, trotters and also wastewater).
Response: The repitiiton has been removed. The only sentence left there is, 'During outbreaks and rumors of outbreaks, cheaper prices are offered for sick and recumbent pigs.'
Q3: Lines 428-432 should also belong to the previous paragraph (iii).
Response: This paragraph is not revised and merged with section iii.
